# Porous Co-Pt Nanoalloys for Production of Carbon Nanofibers and Composites

**DOI:** 10.3390/ma15217456

**Published:** 2022-10-24

**Authors:** Sofya D. Afonnikova, Anton A. Popov, Yury I. Bauman, Pavel E. Plyusnin, Ilya V. Mishakov, Mikhail V. Trenikhin, Yury V. Shubin, Aleksey A. Vedyagin, Sergey V. Korenev

**Affiliations:** 1Boreskov Institute of Catalysis SB RAS, 5 Lavrentyev Ave., 630090 Novosibirsk, Russia; 2Nikolaev Institute of Inorganic Chemistry of SB RAS, 3 Lavrentyev Ave., 630090 Novosibirsk, Russia; 3Center of New Chemical Technologies BIC SB RAS, 54 Neftezavodskaya St., 644060 Omsk, Russia

**Keywords:** cobalt, platinum, porous nanoalloys, X-ray diffraction, alloy catalysts, carbon nanofibers

## Abstract

The controllable synthesis of carbon nanofibers (CNF) and composites based on CNF (Metals/CNF) is of particular interest. In the present work, the samples of CNF were produced via ethylene decomposition over Co-Pt (0–100 at.% Pt) microdispersed alloys prepared by a reductive thermolysis of multicomponent precursors. XRD analysis showed that the crystal structure of alloys in the composition range of 5–35 at.% Pt corresponds to a *fcc* lattice based on cobalt (*Fm-*3*m*), while the CoPt (50 at.% Pt) and CoPt_3_ (75 at.% Pt) samples are intermetallics with the structure *P*4/*mmm* and *Pm*-3*m*, respectively. The microstructure of the alloys is represented by agglomerates of polycrystalline particles (50–150 nm) interconnected by the filaments. The impact of Pt content in the Co_1−x_Pt_x_ samples on their activity in CNF production was revealed. The interaction of alloys with ethylene is accompanied by the generation of active particles on which the growth of nanofibers occurs. Plane Co showed low productivity (~5.5 g/g_cat_), while Pt itself exhibited no activity at all. The addition of 15–25 at.% Pt to cobalt catalyst leads to an increase in activity by 3–5 times. The maximum yield of CNF reached 40 g/g_cat_ for Co_0.75_Pt_0.25_ sample. The local composition of the active alloyed particles and the structural features of CNF were explored.

## 1. Introduction

The production of composites based on carbon nanomaterials (CNMs) attracts great interest due to the continuous expansion of their practical application. In particular, much attention of researchers is focused on the development of metal-carbon M/CNM composites, which find application in various electrochemical reactions [1,2,3,4], catalytic processes [5,6,7] and adsorption [8,9]. Noble metals, such as Pt, Pd, Rh, and Ir, are often used as an active component within the composition of catalysts. Currently, the search for the ways of complete or partial replacement of such expensive metals with much cheaper analogues (Co, Ni, etc.) seems to be reasonable. At the same time, such aspects as dispersion of active particles, uniformity of their distribution within the carbon matrix, as well as structural and textural properties of CNM itself (specific surface area, bulk density, etc.) are important for the target synthesis of M/CNM composites.

The impregnation of the carbon support is the most often used method to prepare the M/CNM composites [4,10,11,12]. To obtain alloy particles consisting of several metals (M_1_, M_2_, etc.), the sequential impregnation of the support with solutions of the precursor salts can be used. However, this approach might lead to an irregular distribution of metals in the alloy particle’s composition. A wide range of carbon nanomaterials, such as graphene [13], fullerenes [14], carbon nanofibers (CNFs) [1,2,3,4,5,7,10,11,12], and carbon nanotubes (CNTs) [15], could be used as a support. Nevertheless, a common disadvantage of such supported composites and catalysts is their susceptibility to rapid deactivation due to mechanical entrainment, washout, or migration of active component particles over the support’s surface with a subsequent loss of dispersion (agglomeration).

The catalytic pyrolysis of hydrocarbons (or catalytic chemical vapor deposition, CCVD) can serve as an appropriate platform for the synthesis of M/CNM composites. Iron subgroup metals (Fe, Co, Ni) are the most commonly used catalysts for CNM synthesis, as well as their numerous alloys with other metals [16,17,18,19]. Among the metals of the iron triad, cobalt is probably the least studied. It is well known that the Co-based composite materials are of a great demand in different areas of catalysis and materials sciences. In particular, composite catalysts containing cobalt and its alloys are applied in such processes as Fischer–Tropsch synthesis [20,21], low-temperature CO oxidation [22], hydrogenation and dehydrogenation of organic substrates [23,24]. At the same time, it should be noted that, despite a fairly uniform distribution of particles within the structure of carbon support, cobalt-carbon composite catalysts can be deactivated due to sintering, surface coking, or encapsulation of the active particles in CNT channels [25].

The relatively new route based on the phenomenon of carbon erosion (CE) of bulk metals and alloys can be proposed as an alternative approach to the preparation of cobalt-carbon (Co/CNM) composites. In the course of the CE process, the spontaneous disintegration of a bulk alloy exposed to a carbon-containing atmosphere occurs. The fragmentation of bulk metal is accompanied by the emergence of active metal particles on which the graphite-like filaments grow [26,27,28]. As a result of the alloy disintegration, a multitude of similar sized active particles directly embedded into the structure of growing carbon nanofibers is formed. The disperse metallic crystals anchored within the bodies of CNF turned to be mechanically separated, which makes them absolutely resistant to migration and inevitable sintering. A significant advantage of CNF-based composites is in stabilization of Co-M alloy particles in the metallic state due to the tight contact with the carbon graphitic matrix, which prevents such catalyst from deactivation caused by chemical modification of active component (e.g., oxidation). The described effect was previously demonstrated for the case of carbon–carbon hierarchical Co-Cu/CNF/(Carbon Cloth) composites in which the fixed Co-Cu particles showed a higher productivity and stability in the ethanol dehydrogenation reaction [29].

It is known that the addition of such metals as Ag, Cu, Sn, and Pt might exert a considerable promoting and stabilizing effect on the performance of cobalt catalyst in the process of hydrocarbons decomposition to produce CNM [30,31,32,33]. In the case of modification with platinum, for instance, the resulting Co-Pt/CNM composite can be used as a catalyst for electrocatalytic reactions (similar to Pt/CNM catalysts), which would lead to a significant savings due to lower cost. In turn, the bulk Co-Pt alloy can be obtained using the previously developed method of a reductive thermolysis of multicomponent precursor salts [33]. This approach makes it possible to obtain single-phase porous nanoalloys with a developed specific surface area, which ensures their ability to interact rapidly with a carbon-containing gas and undergo disintegration with the subsequent growth of CNF. Similar nickel-based alloys have already exhibited their efficiency in the decomposition of various chlorinated hydrocarbons to produce CNFs [26,28].

Thus, the purpose of the present research was to investigate in detail the opportunity to produce Co-Pt/CNF composite materials based on spontaneous fragmentation of bulk Co-Pt nanoalloys used as catalysts. The phase composition and the structure of the obtained Co-Pt alloys were studied by X-ray diffraction analysis and electron microscopy techniques. The synthesized Co_1−x_Pt_x_ alloys of wide composition (x = 0.0–1.0) were subjected to carbon erosion in the course of catalytic pyrolysis of ethylene resulting in the growth of CNF serving as a carbon support. The influence of the Pt concentration in the alloy composition on the catalytic activity of Co in the CNF synthesis was explored. The initial stages of the formation of active particles were studied in detail by SEM, TEM, and XRD methods, and the local elemental composition of the obtained metallic particles (EDX) embedded in the structure of carbon nanofibers was determined.

## 2. Materials and Methods

### 2.1. Synthesis of Co-Pt Alloys

Co-Pt alloys were synthesized, as described elsewhere [33]. The complex salts [Co(NH_3_)_6_]Cl_3_ (Reachem, Moscow, Russia), [Pt(NH_3_)_4_]Cl_2_·H_2_O (Aurat, Moscow, Russia), and acetone (puriss.) were used without further purification. In the typical synthesis, 6 mL of aqueous solution containing [Co(NH_3_)_6_]Cl_3_ (0.223 g, 0.8 mmol) and [Pt(NH_3_)_4_]Cl_2_·H_2_O (0.293 g, 0.8 mmol) were added to 120 mL of cold acetone (~5 °C) with vigorous stirring. The precipitate was collected on a glass filter and dried in the air. The product weight was 0.490 g (95%). Then, the reductive thermolysis of the sample was carried out at 600 °C for 1 h in a hydrogen atmosphere. The thermolysis temperature was chosen based on the in situ XRD study of Co-Pt alloy formation performed in previous research [33]. Afterwards, the sample was cooled in a helium flow to room temperature.

### 2.2. Synthesis of Co-Pt/CNF Composites

Catalytic testing of the porous nanoalloy samples was carried out in a flow-through quartz horizontal reactor. Four specimens of the catalysts, 30.00 ± 0.02 mg each, were located one by one over a quartz plate. The plate was then placed inside the reactor installed in the high-temperature furnace (Zhengzhou Brother Furnace Co., Ltd., Zhengzhou, Henan, China), which is characterized by a stable temperature profile along the reactor length (±5 °C). Then, the samples were heated in an inert atmosphere (Ar) up to reaction temperature (600 °C). The heating rate was 10 °C/min. Next, H_2_ was fed for 15 min in order to remove the oxide film from the surface of the alloy samples. Then, a gas mixture containing high-purity 40 vol.% ethylene (Nizhnekamskneftekhim, Nizhnekamsk, Russia), 20 vol.% hydrogen, and 40 vol.% argon was purged through the reactor at the same temperature. The flow rate of the reaction mixture was 54 L/h. At the end of the reaction, the obtained carbon product was cooled in an argon flow and unloaded from the reactor. After weighing the product, the carbon yield (g/g_cat_) and the bulk density (g/L) of the resulting composite samples were calculated. The experiments were repeated several times to refine the data on the catalytic activity of the samples. The measurement error of the carbon yield was 10%.

### 2.3. Characterization of Co-Pt Alloys and Co-Pt/CNF Composites

The elemental analysis of the samples was performed by atomic emission spectrometry (AES) on a Thermo Scientific iCAP-6500 spectrometer (Thermo Scientific, Waltham, MA, USA). The sample was dissolved in aqua regia at heating and then evaporated with hydrochloric acid up to the complete removal of nitric acid. The relative standard deviation of the Co and Pt determination was 0.03.

X-ray powder diffraction (XRD) analysis of the alloy samples and the obtained carbon product was carried out on a Shimadzu XRD-7000 diffractometer (CuKα- radiation, graphite monochromator, 2Θ angles range of 20–100°, step 0.05°) (Shimadzu, Tokyo, Japan). The phases were identified by comparing the positions and intensities of the diffraction peaks with the PDF-2 data base [34]. Determination of the unit cell parameters was performed by the full-profile method using the PowderCell 2.4 program [35].

The microstructure of carbon product was studied by transmission electron microscopy (TEM) using a JEOL JEM 2100 microscope (JEOL Ltd., Tokyo, Japan) equipped with an energy dispersive X-ray spectrometer INCA-250 “Oxford Instruments” (Oxford Instruments, Abingdon, Oxfordshire, UK). The accelerating voltage was 200 kV, and the resolution was 0.14 nm. The samples were deposited on perforated carbon films fixed on a copper grid. An image of the crystal lattice of gold monocrystals with the Miller index (111)–0.235 nm was used as a standard for the linear size calibration on electron microscopic images. Computer processing of the obtained electron microscopic images was carried out using the Digital Micrograph “Gatan” program, as well as the FFT technique [36,37].

The alloys and the obtained carbon materials were studied by scanning electron microscopy (SEM) using a JSM-5100LV microscope (JEOL Ltd., Tokyo, Japan) equipped with an EX-23000BU EDX spectrometer (SPECTRO Analytical Instruments, Kleve, Germany). The operating voltage of the microscope was 15 kV, the magnification was in a range of 1000–100,000×.

The textural characteristics of the alloys and the carbon nanomaterials were determined using a Sorbtometer analyzer (PromEnergoLab, Moscow, Russia) by the Brunauer, Emmet, and Teller (BET) method. Nitrogen was used as an adsorbate. Before analysis, the samples were degassed in a helium flow at 150 °C for 10 min.

## 3. Results and Discussion

### 3.1. Characterization of Co-Pt Alloys

A series of Co-Pt alloys with a wide range of Pt concentration (0–100 at.%) was synthesized at the first stage of this work. Typical diffraction patterns of the obtained Co-Pt samples are shown in Figure 1. The diffraction patterns of the samples containing 15 and 35 at.% Pt exhibit a set of reflexes corresponding to the *fcc* phase based on the cubic modification of cobalt (*Fm*3-*m,* ICDD PDF-2 #15-0806), as seen in Figure 1. The positions of the characteristic peaks are shifted with respect to the reflexes for pure Co and Pt, thus indicating the formation of Co_1-x_Pt_x_ solid solutions. It should be noted that, besides the main set of reflexes attributed to the *fcc-*phase, there are also the low-intensity peaks at 2Θ ~ 41 and 46° observed on the diffraction patterns of the samples with the Pt content up to 15 at.% (Figure 1). These peaks are most likely related to the solid solution based on the hexagonal (*hcp)* modification of cobalt (*P*6_3_*/mmc,* ICDD PDF-2 #05-0727).

Further increase in Pt concentration (>15 at.%) results in principle changes of the diffraction profiles. As follows from the XRD data, the Co-Pt sample (50 at.% Pt) is composed of a single tetragonal phase with a CoPt intermetallic structure (*P4/mmm*, ICDD PDF-2 #65-8969). The sample with a Pt content of 75 at.% is represented by the single-phase CoPt_3_ intermetallic structure with a primitive cubic crystal lattice (*Pm3m*, ICDD PDF-2 #29-0499). The analysis of XRD data makes it possible to conclude that the phase composition of the synthesized samples corresponds well to the structure of the phase diagram known as the {Co-Pt} binary system [38].

The results of the SEM technique combined with the EDX mapping demonstrate a rather uniform distribution of cobalt and platinum atoms within the obtained alloys (Figure 2). The chemical composition of the synthesized samples determined by the AES method was found to be in good agreement with a target ratio Co/Pt specified during the synthesis (Table 1).

The as-prepared Co-Pt alloys are microdispersed powders of dark grey color. Figure 3 demonstrates the SEM data for the Co-Pt sample with different content of Pt (25, 50, and 75 at.%). It is clear from the SEM image taken at a low magnification (Figure 3a) that the Co-Pt alloys are represented by agglomerates of different size. There are both the small (<10 μm) and large (>100 μm) particles. Further investigation of the secondary structure of the samples by SEM showed that these agglomerates have pores (and possess a sponge-like structure consisting of polycrystalline fragments (50–150 nm in size) interconnected by crosspieces. It is possible to see from the images presented in Figure 3c–h that there are no principle differences in the secondary structure for the Co-Pt alloys with different concentration of Pt dopant. Nevertheless, the sample with high concentration of Pt (75 at.) is seen to have the thinnest structure and smallest size of structural elements (Figure 3g,h). It is worth noting that such a morphology is rather typical for the alloys obtained by the reductive thermolysis of multicomponent precursors [26,28].

Selected TEM images of the bimetallic samples are presented separately for a better demonstration of their primary structure (Figure 4a–c). As such, individual particles are poorly pronounced; only the individual irregularly shaped fragments can be seen. The size of the fragments does not exceed 150 nm due to the specific synthesis procedure. As it was shown earlier, the size of the crosspieces is determined by the temperature and the duration of the reducing stage of thermolysis [39]. It is also possible to observe the presence of a thin layer on the surface of the alloys, which presumably belongs to the oxide shell (Figure 4b–e), resulting from the interaction of the metallic surface with an atmospheric air. It should be noted that the thickness of this layer becomes smaller with an increase in Pt content, which is a well-known effect for alloys in general. Thus, the addition of an inert component to the composition of the alloy increases its resistance to oxidation [40]. The higher the content of the noble metal (Pt) in the alloy, the more resistant the Co-Pt alloy to oxidation. One can see from Figure 4d that the oxide layer on the alloy surface is practically absent in case of the CoPt_3_ sample (75 at.% Pt). The specific surface area of the alloy samples determined by the BET method is varied within a range of 5–10 m^2^/g. The porous structure along with the rather developed surface area makes such alloys more reactive and “qualified” for the process of carbon erosion during the contact with hydrocarbons (ethylene).

### 3.2. Catalytic Decomposition of Ethylene on Co-Pt Alloys

The obtained samples of Co-Pt alloys are not catalysts themselves but play the role of their precursors. The interaction of the alloys with the carbon-containing atmosphere at high temperature (400–800 °C) results in their disintegration and consequently leads to an appearance of the active particles catalyzing the growth of CNF. The process of CE of the alloys occurs very rapidly due to the developed surface of the initial Co-Pt alloys referred to as catalyst precursors.

Figure 5 shows the methodology of catalytic tests carried out in a laboratory reactor. Decomposition of ethylene on a metallic catalyst with formation of CNF proceeds according to the reaction:C_2_H_4_ → 2C (CNF) + 2H_2_

During the experiment, an additional hydrogen is fed along with C_2_H_4_ in order to accelerate the CE process [41].

The presented testing procedure allows one to acquire the catalytic data for several samples at the same time during the single experiment, which makes it possible to rank the samples according to their catalyst performance. The results of the catalytic tests are summarized in the diagram in terms of their productivity (Figure 6). It shows the amount of carbon product (g) grown per 1 g of the alloy during the experiment of 30 min.

It can be seen that the reference samples (100% Co and 100% Pt) are characterized by a very low productivity. For the pure Co-catalyst, the yield of CNF does not exceed the value of 5 g/g_cat_, which is most likely associated with a rapid deactivation of cobalt particles due to their encapsulation with amorphous carbon [42]. It is also clear that the Pt itself (which is supposed to play the role of an activating additive [43]) does not show any activity at all (yield of carbon ~0 g/g_cat_). This result is quite expected since pure Pt does not belong to the family of metals capable of catalyzing the growth of carbon nanostructures [27].

The analysis of the catalytic testing results revealed that the Pt concentration range of 15–50 at.% in Co-Pt alloy seems to be the most interesting from a practical point of view. It can be seen that inside this interval, there is a significant increase in the catalytic performance of the system. The addition of Pt in the amount of 15 at.% leads to a significant increase in carbon yield if compared with the reference Co (Figure 6). A further increase in Pt concentration gives even greater gain in productivity. The sample of Co_0.75_Pt_0.25_ alloy was found to exhibit the highest productivity among the tested samples (40 g/g_cat_). Thus, as a result, it is possible to obtain a composite with the content of Co-Pt active particles ranging from 2.5 to 10 wt%.

At the same time, a continuous increase in Pt concentration (over 25 at.%) results in a drastic worsening of the catalytic performance. For example, the sample with an equimolar ratio of Co and Pt (50/50) shows the same level of productivity as the reference Co sample (< 10 g/g_cat_). Further increase in Pt content up to 75 at.% turns the Co-Pt catalyst to almost inactive form.

When comparing the obtained results with the literature data, it can be concluded that the catalytic behavior of the Co-Pt samples resembles that of the Ni-Pt system previously studied in decomposition of chlorinated hydrocarbons [44]. In particular, it was found that the performance of Ni-Pt catalysts depending on the Pt content passes through a maximum of 27 g/g_cat_ at 4.3 wt % Pt [44]. This can be explained by a dual action of Pt. On the one hand, Pt enhances the catalytic ability of Co to decompose hydrocarbons. On the other hand, the presence of platinum in high concentration inhibits further diffusion of carbon atoms through the bulk of the catalytic Co-Pt particles.

### 3.3. Study of the Carbon Erosion Process of Co-Pt Alloys

The samples of the obtained composite were characterized by XRD, SEM and TEM methods. Figure 7 shows the change in the phase composition of the Co_0.85_Pt_0.15_ sample in the course of reaction (before the interaction with the reaction medium, as well as after 6 and 30 min of reaction). In comparison with the pristine sample (diffraction pattern 1), the reflections at 2Θ = 26.2°, 44.7°, 54.0°, 77.0° appeared in pattern 2, corresponding to the family of graphite crystal reflecting planes (002), (101), (004), (110) correspondingly. The marked reflections become more intensive (pattern 3) as the carbon content in the sample increases. At the same time, a decrease in the intensities of the reflexes related to the initial alloy can be observed.

It is obvious from the SEM data that the interaction of Co-Pt alloys with the reaction mixture is accompanied by their fragmentation (Figure 3 and Figure 4), which is accomplished by the emergence of numerous dispersed particles (Figure 8a,b) playing the role of catalytically active centers for the CNF synthesis. No significant effect of the Pt content on the thickness of the obtained carbon filaments (200–500 nm) was revealed. However, in the case of pure Co, all the grown carbon filaments appeared to be very short, and their average length did not exceed 2 μm. It can be seen that all the observed catalytic particles are covered with a layer of carbonaceous deposits completely preventing them from further interaction with ethylene. Thus, low productivity of pure cobalt should be explained by a rapid deactivation of the as-formed active particles (Figure 6). In Figure 9a,b one can see the TEM images showing the characteristic appearance of the particles responsible for the growth of carbon nanofibers. The observed data also confirm that the active particles are rather quickly “shrouded” by the carbon deposits, which eventually leads to their deactivation.

The result of the interaction of pure 100% Pt sample with ethylene is presented in Figure 8c,d. As follows from the SEM data, there is no evidence of the sample’s destruction forced by the CE process. After being exposed to 30 min of contact with the reaction mixture, the sample is seen to be covered by a thin layer of non-catalytic carbon, fully retaining its original morphology.

The general view and the structure of CNF product obtained over the most active Co-Pt alloy samples are presented in Figure 10. It can be seen that a classical carbon nanomaterial of a filamentous morphology is formed in this case. It should be stressed that the filamentous morphology of carbon product is predominant. In contrast to the reference Co sample, the catalyst promoted by the addition of Pt produces significantly longer carbon filaments (Figure 10a,b). With increasing Pt concentration, the resulting product is represented by very long and less tangled carbon nanofibers (Figure 10c,d). According to the microscopic data taken in the back-scattered electron mode (Figure 10e,f), it is clear that the metallic particles are embedded to the structure of carbon filaments. The majority of particles generate the CNF growth in two opposite directions.

Examination of the obtained composites under transmission microscope reveals that the surface of the active particles is open and free from amorphous carbon deposits (Figure 11a–d). Thus, the proposed method makes it possible to synthesize in one step the composite systems containing Co-Pt alloy particles distributed within the structure of the carbon support. It is important to note that the peculiarity of this approach is the use of a bulk alloy that is capable of growing the carbon product, which further bears the function of a support for the active particles. As a result, one can get a nano-dispersed fixed catalyst where Co-Pt alloy particles of desired composition are separated from each other and physically incapable of agglomeration. Such properties make this material attractive for a wide range of applications.

Then, the character of the Pt distribution throughout the obtained composite (as well as in the initial alloy) was explored and compared. As seen from the EDX data obtained for a number of samples, both metals are evenly distributed in the structure of alloyed particles. Hence, there is no redistribution of metal atoms during the process of carbon erosion of Co_0.75_Pt_0.25_ alloy, which agrees well with the XRD data. The formed active particles are characterized by the chemical composition similar to that of the starting alloy. This allows one to assert that Pt is not inclined to redistribution during the CE process, in contrast to, for example, Cr and Mo present in Ni-Cr and Ni-Mo alloys [45,46].

Figure 12 demonstrates the TEM-EDX data, according to which platinum and cobalt atoms are both localized within the active particles. This fact is also confirmed by the EDX results for the active particles obtained in course of the interaction of CoPt (50 at.% Pt) with ethylene for 30 min (Table 2). Taking into account the experimental error of the EDX method, one can conclude that the ratio of Co/Pt remains the same as for the initial alloy (close to 50/50).

## 4. Conclusions

The method proposed in the present work permits one to synthesize a number of single-phase Co-Pt alloys serving as catalyst precursors. The influence of Pt concentration on the ability of Co-Pt alloys to disintegrate under the action of CE under the reaction conditions of ethylene decomposition was established. The addition of Pt in an amount of 15–50 at.% was found to boost the catalytic performance of Co with respect to CNF synthesis (increase of ~8 times). As a result, the Co-Pt/CNF composites containing metallic particles in a concentration of 2.5–10 wt% can be produced. The ratio of Co and Pt in the obtained fixed particles correspond to the value that was present during the synthesis of the initial alloys. The synthesized Co-Pt/CNF composite materials attract great interest for application in the area of heterogeneous catalysis, including electrocatalytic applications. In particular, this approach makes it possible to reduce the Pt content by replacing it with Co and thus reducing the cost of the Pt-based materials.

## Figures and Tables

**Figure 1 materials-15-07456-f001:**
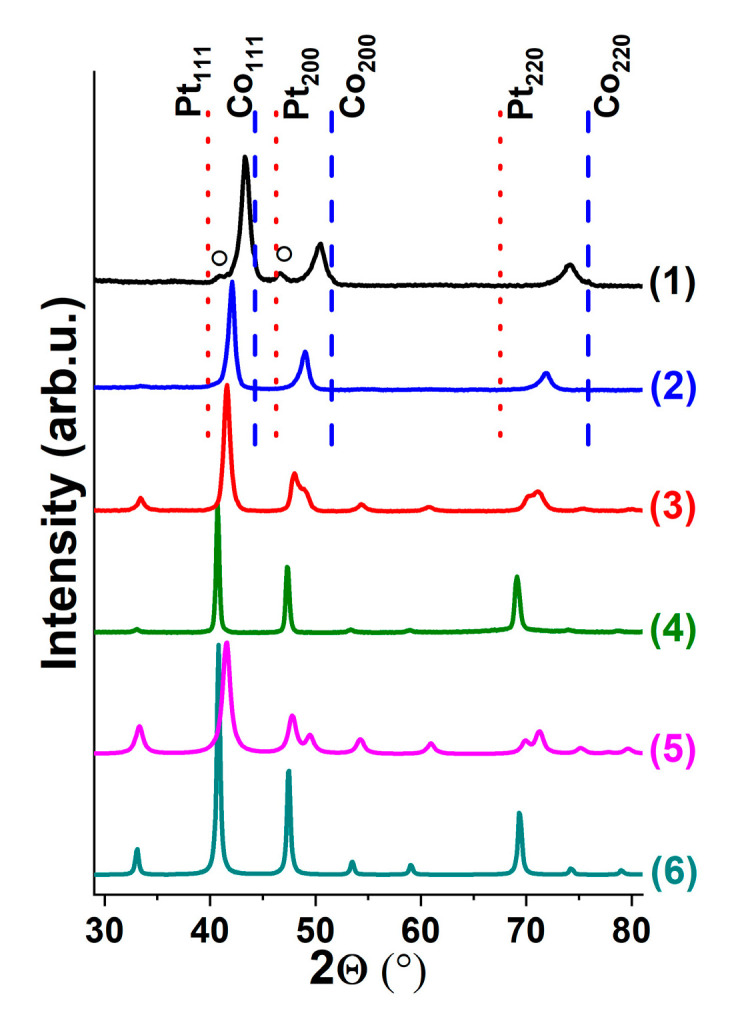
XRD patterns of Co-Pt samples: 1—15 at.% Pt; 2—35 at.% Pt; 3—50 at.% Pt; 4—75 at.% Pt; and the references for ordered phases: 5—CoPt (ICDD PDF-2 #65-8969); 6—CoPt_3_ (ICDD PDF-2 #29-0499). °—reflexes of solid solution based on hexagonal modification of cobalt. The position of reflexes for 100 Co and 100 Pt are shown by dash lines for comparison.

**Figure 2 materials-15-07456-f002:**
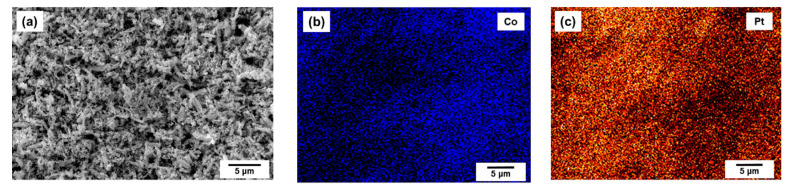
SEM micrograph and EDX element mapping for CoPt (50 at. % Pt) alloy. (**a**)—SEM data; (**b**)—EDX mapping of Co; (**c**)—EDX mapping of Pt.

**Figure 3 materials-15-07456-f003:**
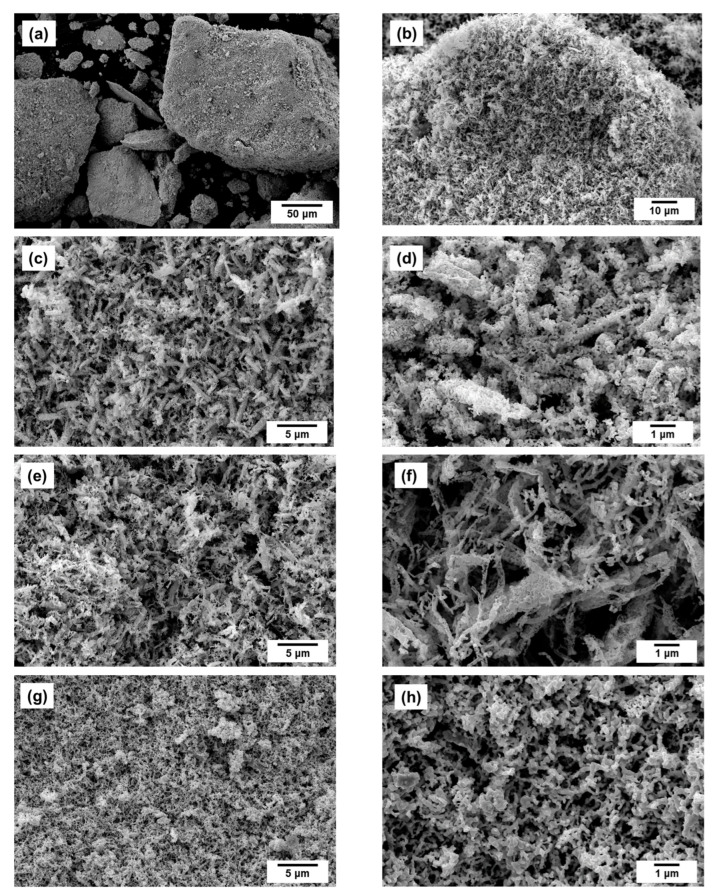
SEM micrographs of the CoPt alloy: (**a**–**d**)—50 at.% Pt; (**e**,**f**)—25 at.% Pt; (**g**,**h**)—75 at.% Pt.

**Figure 4 materials-15-07456-f004:**
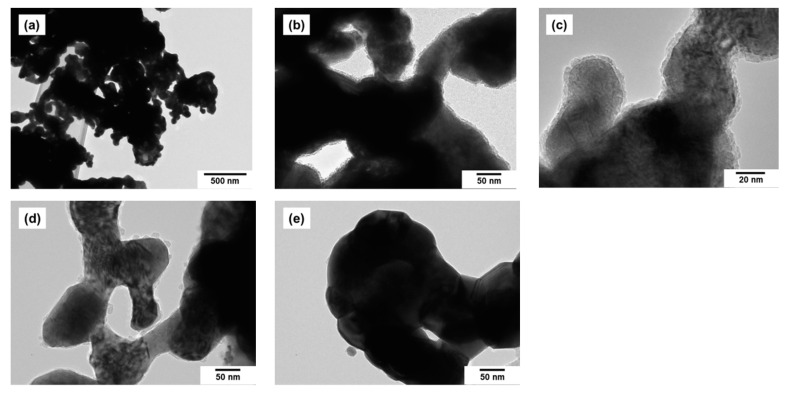
TEM images of Co-Pt alloy samples: 25 at.% Pt (**a**–**c**), 50 at.% Pt (**d**), 75 at.% Pt (**e**).

**Figure 5 materials-15-07456-f005:**
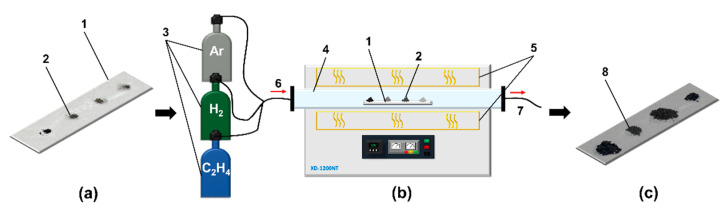
Schematic of catalytic testing of the Co-Pt samples. (**a**)—Co-Pt alloy samples on quartz plate before testing; (**b**)—horizontal flow quartz reactor; (**c**)—CNF samples produced via ethylene decomposition. 1—quartz plate for samples; 2—samples of alloys; 3—gases; 4—quartz reactor; 5—heating element; 6—input of reaction mixture (C_2_H_4_/H_2_/Ar); 7—outlet of gaseous products; 8—obtained samples of composites.

**Figure 6 materials-15-07456-f006:**
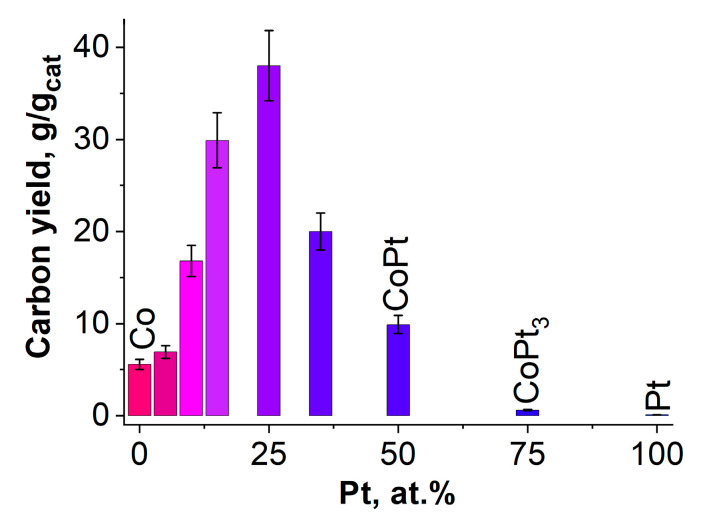
Dependence of the carbon yield on the platinum content in the ethylene decomposition reaction. The data for Co (100%) and Pt (100%) are given for comparison. The reaction conditions are as follows: C_2_H_4_/H_2_/Ar, T = 600 °C, 30 min.

**Figure 7 materials-15-07456-f007:**
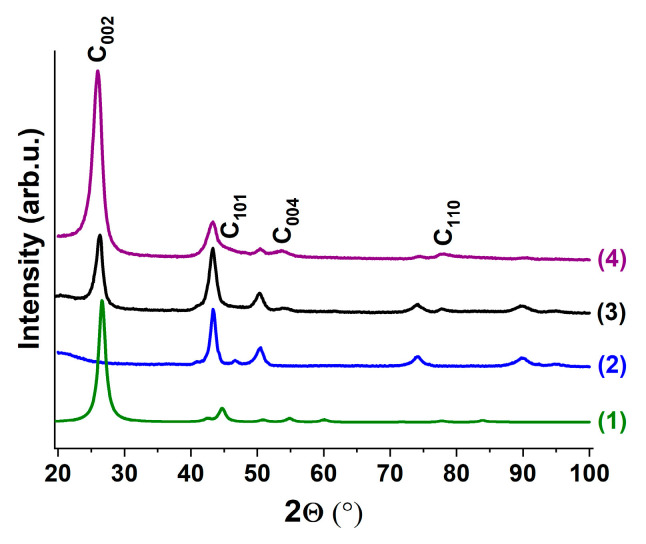
XRD patterns of graphite (reference, ICDD PDF-2 #41-1487) (1), Co_0.85_Pt_0.15_ alloy before (2) and after interaction with reaction mixture: (3)—6 min; (4)—30 min. C_2_H_4_/H_2_/Ar, T = 600 °C.

**Figure 8 materials-15-07456-f008:**
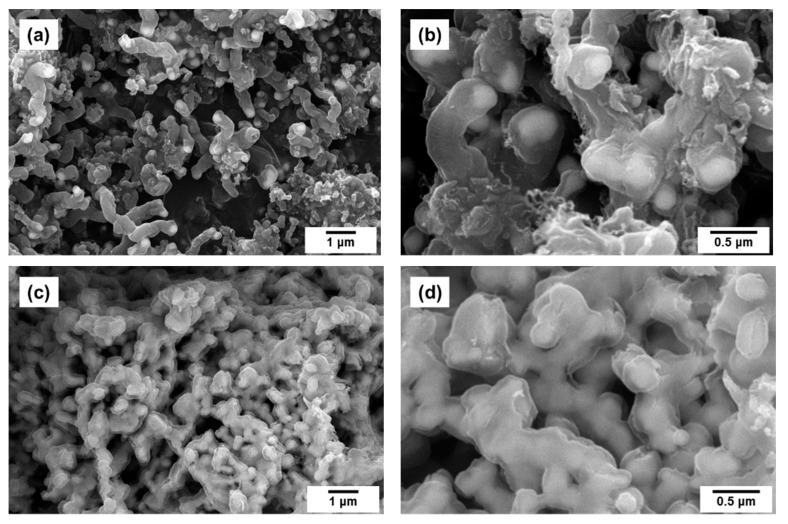
SEM micrographs for Co (100%) (**a**,**b**) and Pt (100%) (**c**,**d**) reference samples after being exposed to reaction mixture (C_2_H_4_/H_2_/Ar) for 30 min.

**Figure 9 materials-15-07456-f009:**
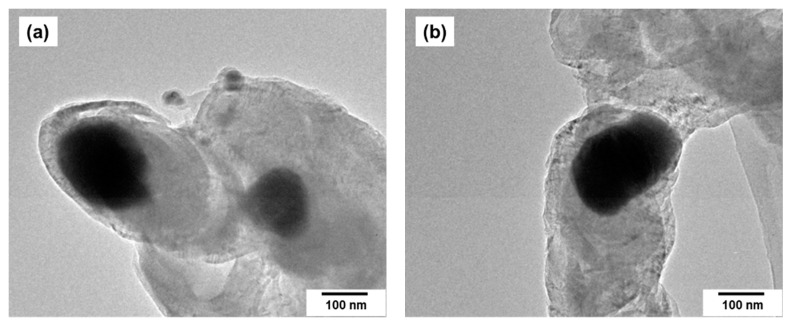
TEM data for the comparison sample Co (100%). The reaction conditions are as follows: C_2_H_4_/H_2_/Ar, T = 600 °C, 30 min, (**a,b**)—active particles.

**Figure 10 materials-15-07456-f010:**
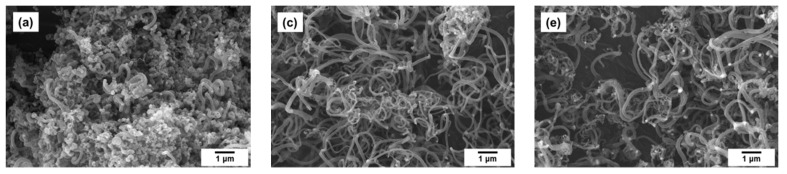
SEM micrographs of composites obtained on Co-Pt alloys: 25 at.% Pt (**a**,**b**), 50 at.% Pt (**c**,**d**), 75 at.% Pt (**e**,**f**) back-scattered electron beam mode. The reaction conditions are as follows: C_2_H_4_/H_2_/Ar, T = 600 °C, 30 min.

**Figure 11 materials-15-07456-f011:**
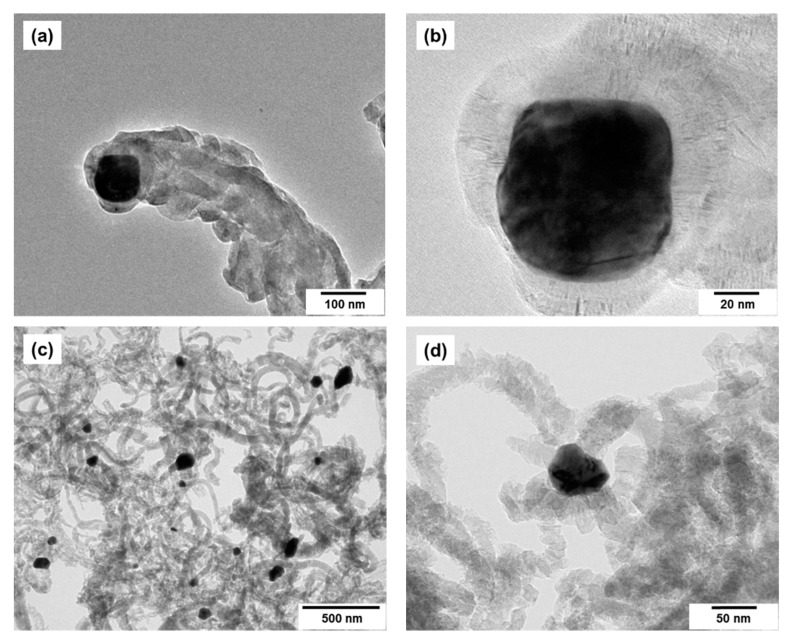
TEM images of Co-Pt/CNF composite samples: (**a**,**b**)—25 at.% Pt; (**c**,**d**)—35 at.% Pt. The reaction conditions are as follows: C_2_H_4_/H_2_/Ar, T = 600 °C, 30 min.

**Figure 12 materials-15-07456-f012:**
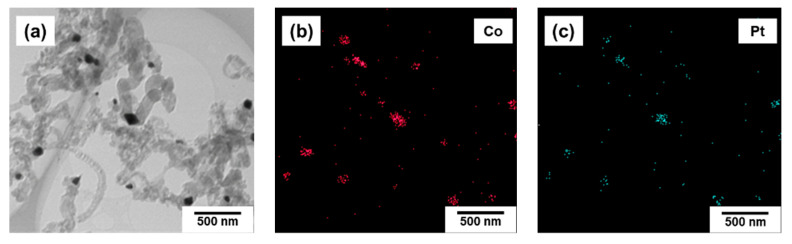
EDX data for Co_0.75_Pt_0.25_ sample after interaction with its reaction mixture (C_2_H_4_/H_2_/Ar, T = 600 °C) for 30 min. (**a**)—image of CNF, (**b**)—distribution of Co atoms (**c**)—distribution of Pt atoms.

**Table 1 materials-15-07456-t001:** Chemical compositions of the synthesized Co-Pt alloys.

Sample	Co_0.95_Pt_0.05_	Co_0.90_Pt_0.10_	Co_0.85_Pt_0.15_	Co_0.75_Pt_0.25_	Co_0.65_Pt_0.35_	CoPt	CoPt_3_
Pt (preset), at.%	5	10	15	25	35	50	75
Pt (AES), at.%	4.4 (3)	9 (1)	14 (1)	23 (2)	34 (3)	47 (3)	73 (5)

**Table 2 materials-15-07456-t002:** EDX data for CoPt (50 at.% Pt) at different areas after being exposed to C_2_H_4_/H_2_/Ar reaction mixture at T = 600 °C for 30 min.

Spectrum	Co, at.%	Pt, at.%
1	46.3	53.7
2	56.7	43.3
3	45.4	54.6
4	52.3	47.7
5	45.4	54.6
Average value	49.2	50.8

## Data Availability

Data are contained within the article.

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
