# Peer review of "Porous Co-Pt Nanoalloys for Production of Carbon Nanofibers and Composites"

_materials, 2022, doi:10.3390/ma15217456_

Round 1
Reviewer 1 Report
The paper presented the catalytic deposition of CNFs with Co-Pt nanoallaoys at 600℃. Both the cystalline structure and morphology of the alloy with different composition of Pt had been fabricated and characterized, as well as their impact on the catalytic activity as the nanocatalyst for the CNF production. Some questions need to be addressed on before being published by the journal. Comments are given below:
1. For the production of the carbon nanomaterial, besides the carbon yield, how about the quality (crystalline structure) of the CNF products? Generally, 1D carbon nanomaterials fabricated at mild temperatures is mainly in the amorphous state, such as those produced by flame deposition method below 500℃ (Surface & Coatings Technology 431 (2022) 128032)
2. Is there any dependence of the CNF thickness on the composition of Co-Pt alloy?
3. The catalytic activity of the alloy could be temperature-dependent. Is there any temperature difference between the four samples on the quartz plate depicted in Fig 5?
Author Response
Dear Reviewer! Thank you very much for your comments and suggestions that allowed us to improve the paper! Below please find the answers to you questions. Q1. For the production of the carbon nanomaterial, besides the carbon yield, how about the quality (crystalline structure) of the CNF products? Generally, 1D carbon nanomaterials fabricated at mild temperatures is mainly in the amorphous state, such as those produced by flame deposition method below 500 C (Surface & Coatings Technology 431 (2022) 128032) A1. Thank you for your question! Indeed, the carbon product synthesized on Co-Pt alloys is characterized by the graphite-like structure, regardless to a specific composition of catalyst sample. The peaks of such a graphite-like phase can be clearly observed in the X-ray diffraction patterns presented for the Co0.85Pt0.15/CNF sample; the positions and intensities of defined peaks almost coincide with those which are characteristic of the graphite peaks. For the readers’ convenience, the diffraction pattern corresponding to the pure graphite phase has been added to Fig. 7 for clarity. Q2. Is there any dependence of the CNF thickness on the composition of Co-Pt alloy? A2. Good question, thanks! Regardless to the Pt content, the thickness of the obtained carbon filaments (200-500 nm) looks quite similar. There is also a certain fraction of tiny nanoparticles (~ 20 nm) resulted from the secondary disintegration. We believe that the study on impact of the Pt content in the composition of Co-Pt alloy on size of the catalytic particles (thickness of carbon filaments), as well as the investigation of its effect upon the morphologic and structural peculiarities of CNF product seems to lay beyond the main focus of the present paper. The comprehensive study of these aspects would be the subject for our research work in the nearest future. Q3. The catalytic activity of the alloy could be temperature-dependent. Is there any temperature difference between the four samples on the quartz plate depicted in Fig 5? A3. In this study, a furnace having a very stable temperature profile along the length of the reactor was used in the catalytic experiments. In fact, for the samples loaded onto the same quartz plate, the temperature variation did not exceed ±5°C. The corresponding information was added to the experimental part of the manuscript. Thank you! On behalf of authors, Sergey V. KorenevReviewer 2 Report
The author synthesized a number of single-phase Co-Pt alloys serving as catalyst precursors. The influence of Pt concentration on the ability of Co-Pt alloys to disintegrate under the action of CE under the reaction conditions of ethylene decomposition was established. However, major revisions are in need to make it more convinced.
- In page 3, please provide the precise amount of [Co(NH3)6]Cl3, [Pt(NH3)4]Cl2 H2O, water, and acetone in the Section “2.1 Synthesis of Co-Pt alloys”.
- In page 3, please offer the heating rate of each thermolysis processes in the Section “2.2. Synthesis of Co-Pt/CNF composites”.
- Why “600︒C” was selected as the thermolysis temperature of Co-Pt alloys and Co-Pt/CNF composites? What about other temperatures?
- It is better to provide hose standard ICDDPDF curves of Co-Pt alloys in Figure 1 to more intuitively distinguish the phase transformation of Co-Pt samples for readers.
- In line 185 of page 4, the author claimed that “The chemical composition of the synthesized samples determined by the AES method was found to be in good agreement with a target ratio Co/Pt specified during the synthesis.” Neither AES data, nor designed Co/Pt ratio were provided, which should be provided with a table.
- Only the morphology of CoPt alloy (50 at.% Pt) was offered in Figure 2 and 3. To clarify the consistency of morphology, the morphology of other Pt dopant amount should also be provided.
- The “oxide shell” that mentioned in line 209 of page 5 should be verified by SAED, rather than purely observed from TEM to directly determine its chemical composition.
- In line 211 of page 5, the author noticed that “the thickness of this oxide shell becomes smaller with an increase in Pt content”, please explain the reason.
Author Response
Dear Reviewer!
Thank you very much for your comments and suggestions that allowed us to improve the paper!
Below please find the answers to you questions.
Q1. In page 3, please provide the precise amount of [Co(NH3)6]Cl3, [Pt(NH3)4]Cl2 H2O, water, and acetone in the Section “2.1 Synthesis of Co-Pt alloys”.
A1. Done. The Section "2.1 Synthesis of Co-Pt alloys" has been changed accordingly.
Q2. In page 3, please offer the heating rate of each thermolysis processes in the Section “2.2. Synthesis of Co-Pt/CNF composites”.
A2. Done. The information has been added.
Q3. Why “600C” was selected as the thermolysis temperature of Co-Pt alloys and Co-Pt/CNF composites? What about other temperatures?
A3. According to our recent in situ study of Co-Pt alloy formation [1], the thermolysis temperature of 600 °C was chosen as the optimal temperature to obtain the single-phase Co-Pt alloys. The higher thermolysis temperature (T > 600 °C) would lead to a significant sintering of the alloys, with subsequent decrease in their catalytic activity. In turn, the thermolysis temperatures below 600 °C will certainly result in the formation of inhomogeneous alloys.
The corresponding changes have been made in The Section "2.1 Synthesis of Co-Pt alloys".
[1]. Popov, A.A.; Shubin, Y.V. et. al. // Nanotechnology 2020, 31, 495604
Q4. It is better to provide hose standard ICDDPDF curves of Co-Pt alloys in Figure 1 to more intuitively distinguish the phase transformation of Co-Pt samples for readers.
A4. Thank you for the suggestion! The corresponding information was added.
Q5. In line 185 of page 4, the author claimed that “The chemical composition of the synthesized samples determined by the AES method was found to be in good agreement with a target ratio Co/Pt specified during the synthesis.” Neither AES data, nor designed Co/Pt ratio were provided, which should be provided with a table.
A5. Agree, thank you. The corresponding Table has been inserted to the text.
Table 1. Chemical compositions of the synthesized Co-Pt alloys.
|
Sample |
Co0.95Pt0.05 |
Co0.90Pt0.10 |
Co0.85Pt0.15 |
Co0.75Pt0.25 |
Co0.65Pt0.35 |
CoPt |
CoPt3 |
|
Pt (preset), at.% |
5 |
10 |
15 |
25 |
35 |
50 |
75 |
|
Pt (AES), at.% |
4.4(3) |
9(1) |
14(1) |
23(2) |
34(3) |
47(3) |
73(5) |
Q6. Only the morphology of CoPt alloy (50 at.% Pt) was offered in Figure 2 and 3. To clarify the consistency of morphology, the morphology of other Pt dopant amount should also be provided.
A6. Agree. The corresponding SEM micrographs have been added to Figure 3 and discussed in the revised text.
Q7. The “oxide shell” that mentioned in line 209 of page 5 should be verified by SAED, rather than purely observed from TEM to directly determine its chemical composition.
A7. Agree. At the same time, we believe that the mentioned aspect doesn’t seem to be crucial for the principle conclusions made. Since all the prepared Co-Pt samples were contacting with atmospheric air before examination by SEM, it was reasonable to suggest the presence of the oxide shell over the surface of the alloys. The corresponding sentence has been rephrased accordingly.
The dependence of the thickness of this layer upon the Pt content might serve as an indirect confirmation to the appearance of the oxide. It should be though stressed that, in the reaction conditions (600°C, presence of H2) the observed oxide shell has to be rapidly reduced and converted into pure metal, and should not interfere the whole process.
Q8. In line 211 of page 5, the author noticed that “the thickness of this oxide shell becomes smaller with an increase in Pt content”, please explain the reason.
A8. As it was mentioned above, the impact of the Pt content on the thickness of surface layer might serve as an indirect confirmation to the appearance of the oxide over the alloy’s surface. The addition of an inert component to the alloy can significantly increase its resistance to oxidation. Such an effect is well-known and described for the Ni-based alloys as an example [2]. The corresponding amendments were inserted into the text.
[2]. C.W. Corti, D.R. Coupland, and G.L. Selman, Platinum-Enriched Superalloys. Enhanced Oxidation and Corrosion Resistance for Industrial and Aerospace Applications., Platin. Met. Rev., 1980, 24(1), 2–11
Thank you!
On behalf of authors,
Sergey V. Korenev
Round 2
Reviewer 2 Report
The author has thoroughly replied the comments. So, the manuscript can be fully accepted in its current form.